# Can Test-Time Scaling Improve World Foundation Model?

**Wenyan Cong**[1*], **Hanqing Zhu**[1*], **Peihao Wang**[1], **Bangya Liu**[3], **Dejia Xu**[1],
**Kevin Wang**[1], **David Z. Pan**[1], **Yan Wang**[2], **Zhiwen Fan**[1,4], **Zhangyang Wang**[1]
[1]University of Texas at Austin, [2]NVIDIA,
[3]University of Wisconsin-Madison, [4]Texas A&M University
[*]Equal contribution. Correspondence to `atlaswang@utexas.edu`.

## Abstract

World foundation models, which simulate the physical world by predicting future states from current observations and inputs, have become central to many applications in physical intelligence, including autonomous driving and robotics. However, these models require substantial computational resources for pretraining and are further constrained by available data during post-training. As such, scaling computation at test time emerges as both a critical and practical alternative to traditional model enlargement or re-training. In this work, we introduce **SWIFT**, a test-time scaling framework tailored for WFMs. SWIFT integrates our extensible WFM evaluation toolkit with process-level inference strategies, including fast tokenization, probability-based Top-K pruning, and efficient beam search. Empirical results on the COSMOS model demonstrate that test-time scaling exists even in a compute-optimal way. Our findings reveal that test-time scaling laws hold for WFMs and that SWIFT provides a scalable and effective pathway for improving WFM inference without retraining or increasing model size. Project page: https://scalingwfm.github.io/.

## 1 Introduction

World foundation models (WFMs) aim to simulate the dynamics of the physical world by predicting future states based on current observations and inputs. They play a central role in physical intelligence, particularly as a means of generating synthetic data, supporting a wide range of downstream tasks such as autonomous driving, robotics, and embodied planning for scalable simulation, analysis, and agent training.

Despite their promise, training WFMs at scale present significant challenges. Pretraining demands massive computational resources, especially since WFMs operate on video inputs. For example, the most recent COSMOS Agarwal et al. (2025) is trained on tens of millions of video hours using thousands of high-end GPUs over several months. Even post-training, model scaling is often constrained by data availability and diminishing returns from scaling laws. These limitations highlight a critical need for test-time computation scaling—a strategy to improve performance by allocating additional compute during inference, without enlarging or retraining the base model.

Motivated by the success of test-time scaling in large language models and diffusion-based vision-language models Snell et al. (2024); Ma et al. (2025), we, for the first time, explore the test-time scaling for world foundation models.

However, several key challenges make this far from a straightforward borrow-and-test scenario. ①: *We need to have a dedicated evaluation testbed for WFMs.* Existing video generation benchmarks typically emphasize aesthetic quality or semantic alignment, which do not align well with the goals of physical realism and consistency that world modeling requires. Moreover, *a plausible and extensible evaluation toolkit* is needed to assess WFM outputs across diverse downstream applications. In contrast, traditional RL-based reward models are often rigid and difficult to generalize across tasks or domains. ② *A practical and effective test-time scaling strategy must be tailored for WFMs.* Unlike LLMs, WFMs commonly rely on

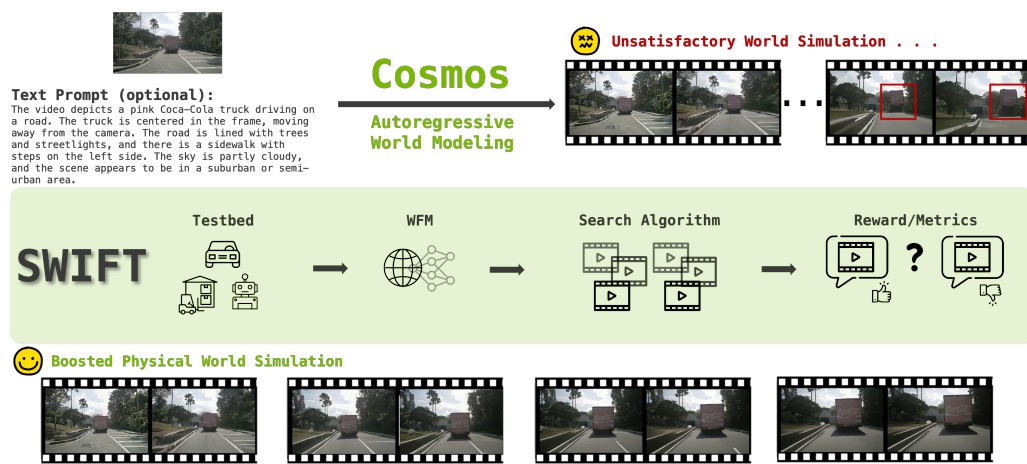

Figure 1: The **SWIFT** pipeline for studying test-time scaling (TTS) in world foundation models (WFMs). We use autoregressive world model COSMOS as a base, which initially produces an unrealistic world simulation (top panel). By applying our TTS method, the simulation is significantly enhanced and becomes more physically plausible (bottom panel).

diffusion-based decoders to produce high-quality outputs, which are inherently slow. This makes it impractical to directly apply advanced test-time strategies like chain-of-thought or tree-of-thoughts, which involve checking intermediate steps.

To address these challenges, we introduce SWIFT, a test-time scaling framework specifically designed for WFMs, as illustrated in Figure 1.

Our contribution is as follows:

- **WFM Evaluation Toolkit**: We propose the first evaluation toolkit specifically designed for world foundation models. The toolkit is modular, extensible, and supports a range of rule-based metrics, enabling multi-aspect evaluation across diverse downstream tasks.

- **WFM Test-Time Scaling Framework**: We introduce **SWIFT**, the first test-time scaling framework tailored for WFMs. SWIFT enables efficient process-level test-time search by incorporating a fast tokenizer for accelerated decision-making, probability-based Top-K pruning to mitigate overconfidence, and a beam search algorithm to maintain performance-efficiency trade-offs.

- **Empirical Findings**: We conduct the first empirical study of test-time scaling for WFMs using COSMOS as a base model. Our findings reveal that: (1) A test-time scaling law exists for WFMs—surprisingly, even under compute-optimal conditions. Small models enhanced with test-time scaling can match or surpass the performance of significantly larger models, given the same inference-time compute budget. (2) Our proposed framework, **SWIFT**, further boosts performance as the number of generated samples increases, and does so efficiently through carefully designed inference-time strategies.

## 2 Motivation: Why Test-Time Scaling For World Foundation Models

*Test-time scaling* has shown promise for leveraging a model's full capacity, where allocating more computation at inference can yield substantial performance gains, even exceeding the benefits of simply increasing model size Snell et al. (2024).

World Foundation Models (WFMs) simulate real-world dynamics and bridge the physical and digital domains, playing a key role in generating domain-specific synthetic data for applications like autonomous driving and robotics. To investigate whether WFMs can similarly benefit from test-time scaling, we highlight two key motivating factors:

① **Training a larger WFM is extremely costly in both compute and data!** Unlike LLMs' training, primarily involving text, WFMs process massive amounts of video data, requiring immense resources. For instance, training the COSMOS autoregressive video model on

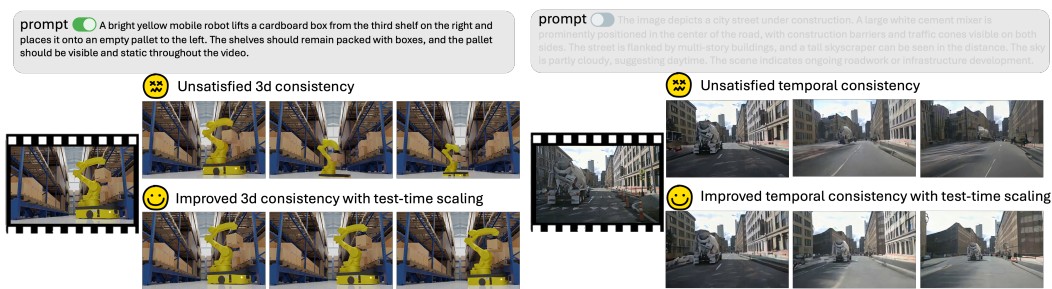

Figure 2: Videos of world foundation model without (top) and with (bottom) TTS, where TTS improves 3D consistency (left) and temporal consistency (right) in the generated videos.

20M hours of footage demanded 10K NVIDIA H100 GPUs over 3 months—even for a 13B model. Moreover, scaling laws suggest that bigger models demand more data. Collecting such large-scale, domain-specific video is substantially hard. This creates a *chicken-and-egg problem*: models intended to generate synthetic video as digital twins require high-quality data—yet such data often only becomes available after a capable WFM has been developed.

② **Larger WFM inference is nearly as costly as running multiple smaller models.** Even after training, inference for WFMs remains a bottleneck. Generating long, high-resolution videos with an autoregressive model is computationally expensive because each step requires a full transformer pass. This process results in slow generation speeds and high memory usage, making large-scale deployment challenging. Under these constraints, test-time scaling—which invests extra computation resources during inference—becomes an attractive alternative to simply building bigger models. For example, running a 12B model requires approximately three times the FLOPs of a 4B model, meaning we can afford to run the 4B model two to three times instead.

**Our Aim: Enhance WFMs at Test Time without Enlarging or Retraining Them.**

# 3 WFM Evaluation Toolkit

World Foundation Models have recently evolved to incorporate video generation as a means of building digital twins—synthetic video representations that simulate the physical world. This capability enables powerful applications in simulation, analysis, and agent training across domains like *autonomous driving* and *robotics*.

Despite this progress, there is still no standardized evaluator tailored to the unique goals of WFMs. ① General video generation benchmarks (e.g., VBench Huang et al. (2024), VideoScore He et al. (2024)) focus on aesthetic or text-conditioned outputs, which are misaligned with the physical realism and dynamical consistency central to world modeling. ② Task-specific benchmarks (e.g., ACT-Bench Arai et al. (2024)) measure downstream control performance but overlook key video generation aspects such as temporal coherence.

To address this gap, we introduce the first general evaluation toolkit for world foundation models, specifically designed to assess their capabilities across diverse domains. Unlike task-specific or generic video metrics, our toolkit defines a modular, extensible framework that supports domain-specific evaluation while remaining broadly applicable.

Our evaluation toolkit now includes the following key metrics:

- **3D Consistency:** Assesses the geometric coherence of the generated scene (left in Figure 2), critical for tasks like autonomous driving. We use the 3D foundation model **CUT3R** Wang et al. (2025), which reconstructs 3D structure from video in a feed-forward manner, enabling efficient evaluation.

- **Temporal Consistency:** Evaluates whether generated video sequences maintain coherent temporal dynamics—preserving smooth background transitions and object permanence across frames, which is essential for tasks such as robotics and planning (right in Figure 2). We measure this using CLIP and DINO similarity scores.

- **Spatial Relationship Awareness:** Measures whether spatial relationships between entities—especially human-environment interactions—are physically plausible. For instance, in factory simulations, we assess adherence to left-right and top-bottom relationships implied by prompts, inspired by Huang et al. (2024).

- **Perceptual Quality:** Assesses the visual fidelity and aesthetic appeal of generated content. Since WFMs must model the world vividly and convincingly, we focus on richness of color, harmony, and artistic quality. We use the LAION aesthetic predictor to evaluate each frame individually. Low-level distortions (e.g., noise or blur) are not penalized, as they can reflect natural properties of real-world sensors.

- **Text-to-Video Alignment:** Measures semantic alignment between the generated video and its associated text prompt. We employ both CLIPScore (frame-level cosine similarity between prompt and visual features) and X-CLIPScore (video-level alignment using cross-modal embeddings).

As noted in the COSMOS work, evaluation of WFMs is inherently challenging due to their generality and complexity. While our proposed toolkit is not exhaustive, it is intentionally modular and extendable, allowing new metrics and tasks to be incorporated as the field matures. To support further research and reproducibility, the toolkit will be open-sourced.

Thoroughly evaluating WFMs across different domains is challenging, as generating thousands of videos demands substantial time and GPU resources. Most existing world modeling work Hu et al. (2023) has focused on autonomous driving, a domain where generating realistic and diverse data—particularly for rare corner cases—is both critical and difficult. In this work, we also adopt autonomous driving as our primary testbed for studying test-time scaling strategies, which aligns with the COSMOS model's target application scenario.

## 4  SWIFT: The First Test-Time Scaling For World Foundation Models

Built upon our evaluation toolkit, we propose **SWIFT**, the first test-time scaling framework for WFMs, as illustrated in Figure 1, to address the following timely and important questions:

**Q① Can test-time scaling improve WFMs even in a compute-optimal way?** Beyond verifying test-time scaling, we also aim to explore if a smaller model, with test-time scaling under a fixed compute budget, can match, or even outperform, a larger model.

**Q② How can we design an effective and practical test-time scaling strategy tailored to WFM video generation?** Rather than directly adapting test-time scaling techniques from LLMs, we seek to design strategies that address the unique challenges of WFMs—such as sequential autoregressive video generation and the high cost of diffusion-based decoding.

### 4.1  Scaling via Test-Time Verifiers: Framework

To demonstrate the effectiveness of test-time scaling for world foundation models, we focus on its autoregressive video generation paradigm. Diffusion-based video generation has been studied extensively for its test-time scaling behaviors (e.g., Ma et al. (2025)), the test-time scaling properties of autoregressive video models remain underexplored. This paradigm is especially relevant, as it offers a unified approach to multimodal generation—recently exemplified by GPT-4o's image capabilities (which is believed to be the underlying recipe).

We first formalize the generation process in autoregressive WFMs. Let $p_\Theta$ denote a pretrained WFM with parameters $\Theta$. This model processes an input consisting of a video chunk $v_0$, and prompt $c$ to generate a video represented as $\mathcal{V} = \{v_1, v_2, \ldots, v_N\}$, where $\mathcal{V}$ consists of $N$-step response. Each step response $v_i$ contains $k$ frames, instead of a single frame, for efficiency and temporal consistency, generated in an autoregressive manner:

$$v_i = p_\Theta(v_i \mid c, v_0, v_1, v_2, \ldots, v_{i-1}). \tag{1}$$

**Formulate the video generation as a Markov Decision Process (MDP):** We draw on prior work that casts sequential decision-making tasks into an MDP framework. Since a WFM generates each step of a video in an autoregressive manner—conditioning on both the input and all previously generated frames—its generation naturally maps to an MDP represented

by the tuple $(\mathcal{S}, \mathcal{A}, \mathcal{R})$. Each state $s \in \mathcal{S}$ corresponds to a partially generated video segment, beginning from an initial state $s_0$ that includes input frames and any textual conditions. The reward function $\mathcal{R}(s, a)$, or called *verifier* Snell et al. (2024), evaluates the quality or alignment of the generated content, while the action space $\mathcal{A}$ dictates how to refine the model's outputs given a search algorithm.

## 4.2 Scaling via Test-Time Verifiers: Verifier $\mathcal{R}$ Design

In the context of test-time scaling, verifiers (rewards) play a central role in assessing whether a generated candidate is desired. Broadly, two categories of reward design exist: **(1) Preference-based rewards** aim to simulate human preferences by leveraging real-world feedback. But collecting large-scale human annotations is labor-intensive, prompting recent efforts to train reward models as human judgments proxies Guo et al. (2025b); **(2) Rule-based rewards** rely on feature-based metrics. Since they do not incorporate external biases, they are generally more objective and free from inductive bias.

**Our Design Choice: Rule-Based Rewards for Robustness and Extensibility:** To determine the more suitable reward formulation for autoregressive WFMs, we conduct a preliminary analysis using a best-of-$N$ sampling strategy. For each prompt, we generate $N$ candidate videos and evaluate them using both reward types. For preference-based rewards, we adopt *VideoScore* He et al. (2024), which provides holistic assessments across multiple aspects (e.g., visual quality, temporal consistency). Given our unconditioned video generation task in autonomous driving using COSMOS-4B, we retain only the relevant dimensions—*visual quality* and *temporal consistency*—and exclude those tied to conditioning signals. For rule-based rewards, we select established metrics corresponding to these same two aspects (e.g., *aesthetic quality* for visual quality and a *object permanence* score for consistency).

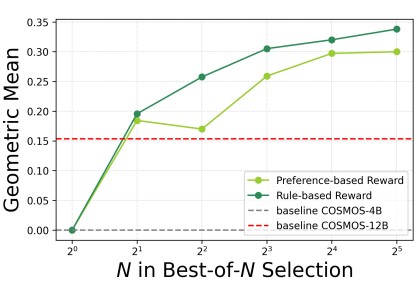

Figure 3: Rule-based vs. preference-based rewards.

As shown in Figure 3, rule-based rewards consistently outperform preference-based ones in both stability and alignment with qualitative inspection. This corroborates findings of *DeepSeek r1* Guo et al. (2025a), which also reported that rule-based evaluation is more robust and less prone to reward hacking. In addition to their reliability, rule-based rewards are easier to extend. *Our WFM evaluation toolkit offers a modular design that allows new metrics to be seamlessly integrated into the verification pipeline, making it scalable for future evaluation needs.*

## 4.3 Scaling via Test-Time Verifiers: Action $\mathcal{A}$ Design

In LLMs, two common strategies are used to improve output quality: (1) modifying the *input prompt* to alter the proposal distribution, and (2) sampling multiple candidate completions and selecting or refining the best one. However, directly adapting strategy (1) to video generation is challenging. First, integrating textual feedback for "reflection" typically requires *a specialized reward model* to detect misalignments and provide correction signals—tools that are not yet available for WFMs. Second, current WFMs are not trained to understand or act on such feedback without *additional fine-tuning*. Therefore, we focus on strategy (2)—*test-time search*—as the primary action to explore test-time scaling laws. We encourage the WFM to explore multiple possible continuations through i.i.d. sampling from its output distribution, following prior work Snell et al. (2024); Guo et al. (2025b). This approach avoids the need to generate reflection-based feedback or train the WFM to interpret and act on such feedback.

Another possible way to introduce randomness in video generation is by keeping the prompt fixed while sampling diverse input frames, potentially using text-to-image or text-to-video models to initialize different starting conditions. However, this strategy is not suitable. First, it introduces inherent variation in the inputs, making the resulting generations not directly comparable. For instance, one input may produce a small object while another generates a larger one, which can significantly affect evaluation metrics such as 3D consistency. Second, WFM may only rely on video as input, as in COSMOS 4B.

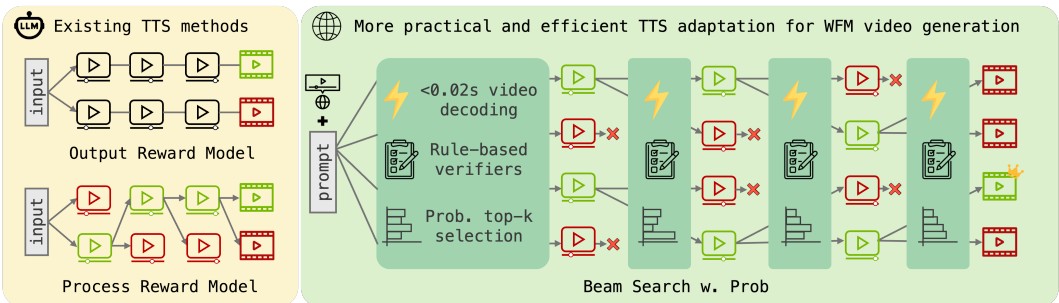

Figure 4: Compared to ORM and PRM, our proposed beam search with probability is more efficient and practical for WFMs with three key efficiency designs.

## 4.4 Scaling via Test-Time Verifiers: Search Algorithm

Within the SWIFT framework, one can seamlessly adapt various test-time scaling methods.

### 4.4.1 Naive best-of-N to verify test-time scaling law

We begin with the simplest possible test-time scaling strategy: a best-of-$N$ search. For each input, we sample $N$ independent video continuations, fully generate each one, score them using our rule-based verifiers $\mathcal{R}$, and then select the highest-scoring video (Figure 4, left). This approach mirrors the "best-of-$N$" output selection commonly used in LLM research (ORM). Although conceptually trivial, best-of-$N$ serves as a crucial start to investigate test-time scaling laws in WFM.

From this experiment, we draw two key observations:

*Observation 1: Test-Time Scaling Exists in WFM*: Even a modest increase in the number of sampled continuations consistently improves generation quality across most evaluated metrics. Simply investing more compute at inference produces higher-fidelity videos, demonstrating that WFMs exhibit a clear test-time scaling law (detailed quantitative results in Figure 3, Table 1, and Table 2).

*Observation 2: Test-Time Scaling Is Surprisingly Compute-Optimal.* Beyond confirming that test-time scaling improves video quality, we ask: how much extra inference compute does a smaller model need to rival a larger one? Strikingly, only two to four additional inference passes of a 4B model match or exceed the output quality of a single pass by a 13B model. In terms of FLOPs, this is comparable to running the larger model once. This highlights a compelling trend in WFMs: test-time scaling can offer a more compute-efficient alternative to model size increases—echoing recent findings in LLMs Snell et al. (2024).

### 4.4.2 Adapting Test-Time Scaling Strategy to World Modeling Practically and Efficiently

Naive best-of-$N$ sampling improves video quality, but it fails to exploit the inherently sequential nature of autoregressive video generation (see our MDP formulation in Sec. 4.1). Intuitively, one can check video quality step-wise and select the best path to generate subsequent frames, similar to the Process Reward Model (PRM) in the LLM Ma et al. (2023).

However, straightforward PRM implementations run into one fatal inefficiency due to:

*Challenge: Expensive decoding via a diffusion-based decoder*: WFMs rely on a diffusion-based denoising tokenizer to produce high-quality frames. This process takes $\approx 137.2$ s on A6000, roughly as long as autoregressively generating those frames. Hence, performing a full diffusion decode step-wise creates substantial overhead. While test-time scaling inherently trades extra compute for better output quality, an efficient strategy must avoid wasted decoding and instead focus compute on exploring promising candidate trajectories.

To address this challenge, we introduce several key contributions to design a practical test-time scaling framework for WFMs, as shown in Figure 4:

**1. Fast-Tokenizer to Accelerate Decision Process** To avoid running the diffusion decoder at every generation step, we leverage the WFM's discrete-token decoder as a lightweight proxy. This "fast tokenizer" converts intermediate latent outputs into video frames in only ≈0.015 s — versus ≈130 s for a full diffusion decode. We find that the reward scores computed from fast-tokenized outputs show strong correlation with those from the diffusion decoder (see Figure 5). This tight alignment confirms that the fast tokenizer offers a reliable and drastically cheaper feedback signal, enabling early pruning of low-quality trajectories and allowing compute to focus on more promising candidate paths.

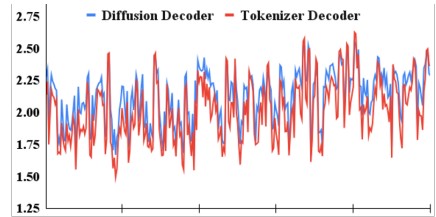

Figure 5: Scores for videos decoded by diffusion decoder (slow) and tokenizer decoder (fast) follow the same trend.

**2. Probability-Based Top-K Pruning for N Samples** Video generation is inherently sequential: early frames influence—but do not guarantee—the quality of later frames. Rather than greedily selecting the single highest-scoring continuation at each step (as in a basic PRM), we maintain a small pool of promising candidates. Concretely, at timestep $t$ we sample $N$ next-step continuations $\{s_t^i\}_{i=1}^N$ and compute their verifier scores $\{r_t^i\}_{i=1}^N$. Instead of deterministically choosing the top-$K$ by score, we perform a probability-based selection using a softmax over verifier scores:

$$p_i = \frac{\exp(r_t^i/\tau)}{\sum_{j=1}^N \exp(r_t^j/\tau)}, \quad \mathcal{B}_{t+1} = \text{SampleTopK}(\{s_t^i\}, \{p_i\}, K),$$

where $\tau$ is a temperature hyperparameter controlling randomness, and SampleTopK draws $K$ unique candidates without replacement. This stochastic top-$K$ strategy introduces controlled exploration, yielding consistently better performance than both top-1 sampling and deterministic top-$K$ sorting (Figure 6).

**3. Beam-Search Algorithm to Maintain Efficiency** To prevent an exponential explosion of candidate trajectories, we adopt a beam-search–inspired procedure that bounds search complexity at each generation step. Specifically, we maintain a fixed beam of $K$ partial video sequences. At each timestep, every beam member spawns $M$ new continuations, yielding $M \times K$ candidates. We score all candidates using our fast tokenizer verifier, then prune back to the top $K$ for the next step. By capping the beam size, this approach keeps branching growth linear in sequence length, sharply reducing wasted computation and ensuring inference cost remains predictable.

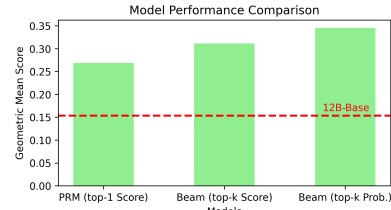

Figure 6: Improvement based on COSMOS-4B using different search algorithms. Beam search with probability boosts the performance most.

## 5 Experiments

**Experimental Settings.** As aforementioned, we use autonomous driving as the test case to demonstrate the test-time scaling on world models. Here we focus on two modalities: image to video, and image+text to video. For autonomous driving, estimating coherent futures requires at least three essential priors that inherently govern the future motion of instances in the scene: position, velocity, and acceleration, as shown in Gao et al. (2024); Hu et al. (2024). So we use the variants that take 9 video frames as input.

**Datasets and Metrics.** We collect a total of 900 input videos from the test splits of two representative autonomous driving datasets: nuScenes and Waymo, each containing 150 scenes. Specifically, we sample one input video from each nuScenes test scene and five inputs from each Waymo test scene. The accompanying text prompts are generated using a prompt upsampling model, which is fine-tuned from the base model in Mistral & NVIDIA.

by the COSMOS framework, ensuring that the prompts are aligned with the style and semantics expected by the world model.

For evaluation, following convention in autonomous driving field, we adopt FID and FVD to assess the overall quality of the generated video frames and sequences, respectively. Note that nuScenes and Waymo are from two data sources thus have different distribution, so we report FVD and FID on each dataset. To provide additional insight, we also evaluate using the VBench and VideoScore (VQ, TC, DD, FC) benchmarks. Since our rule-based rewards partially overlap with the metrics in VBench, we use the remaining two dimensions (motion smoothness (MS), imaging quality (IQ)) from VBench that also focus on temporal consistency to perform cross-validation of our evaluation. To aggregate performance across multiple evaluation dimensions, we compute the geometric mean of the normalized scores, providing a single summary metric that reflects overall quality (as in Figure 3 and Figure 6).

### 5.1 Quantitative Results

**Reward Ablation.** Before conducting our test-time scaling experiments, we perform an ablation study on evaluation metrics to identify which ones are most suitable for use as reward functions in the autonomous driving domain. Using COSMOS-4B as the base model, we apply a naive best-of-N selection strategy and evaluate the final outputs using FVD.

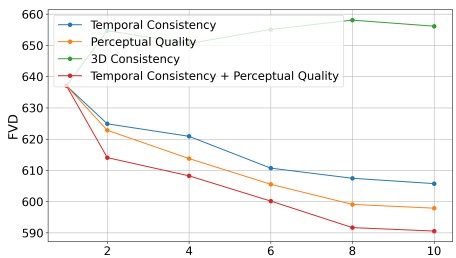

The results, shown in Figure 7, reveal that temporal consistency and perceptual quality contribute most significantly to performance improvements. In contrast, 3D consistency shows limited benefit—likely due to the nature of driving scenes,

Figure 7: COSMOS-4B using different metrics as reward. We use Temporal Consistency and Perceptual Quality by default.

where long-range point clouds introduce noise and reduce the reliability of MSE-based geometric evaluation. We do not consider spatial relationship awareness in this setting since the spatial configurations in driving environments tend to be straightforward (e.g., vehicles aligned along roads), resulting in few violations of spatial constraints implied by the prompt.

| Model | N | FVD | FID | IQ | MS | VQ | TC | DD | FC |
|---|---|---|---|---|---|---|---|---|---|
| 4B | 1 | 637.08/120.30 | 67.75/10.58 | 63.48 | 0.9822 | 3.86 | 3.56 | 3.86 | 3.68 |
| | 2 | 622.82/120.18 | 58.93/10.27 | 63.97 | 0.9831 | 3.86 | 3.56 | 3.86 | 3.68 |
| | 4 | 613.77/117.39 | 52.01/10.25 | 64.33 | 0.9833 | 3.87 | 3.57 | 3.86 | 3.68 |
| | 8 | 599.10/116.14 | 49.27/10.14 | 64.65 | 0.9836 | 3.88 | 3.58 | 3.87 | 3.70 |
| | 16 | 599.08/120.04 | 45.75/10.14 | 64.82 | 0.9839 | 3.90 | 3.59 | 3.89 | 3.73 |
| 12B | 1 | 560.86/117.23 | 67.10/10.67 | 63.73 | 0.9807 | 3.94 | 3.63 | 3.93 | 3.76 |

Table 1: Comparison of COSMOS-4B and COSMOS-12B using naive best-of-*N* sampling. FVD and FID scores are reported separately for the nuScenes (left) and Waymo (right) datasets, due to significant differences in their underlying data distributions.

**Naive Best-of-N.** As shown in Table 1, we begin with a simple best-of-N selection strategy at the output level to demonstrate the effectiveness of test-time scaling. Our key observations are as follows: 1)*Test-time verification improves generation quality.* Compared to the COSMOS-4B baseline, integrating a best-of-*N* (or ORM) as a verifier significantly enhances performance across nearly all evaluation dimensions. 2) *Test-time scaling follows a clear trend.* As N increases, performance improves consistently, indicating a strong scaling law even at inference time. 3) *Compute-optimality at small N.* Notably, a best-of-2 strategy delivers substantial gains without exceeding the inference-time compute budget of a much larger pretrained model (e.g., COSMOS-12B), demonstrating the practical value of lightweight scaling. While smaller models may have a performance gap compared to larger models in approximating the true data distribution, this limitation can be partially mitigated through

effective test-time selection. Moreover, prior work (e.g., Gao et al. (2024)) has shown that automatic metrics such as FVD do not always align well with human-perceived quality. As a result, we place greater emphasis on multi-aspect evaluation across diverse benchmarks. A model is considered improved when it shows consistent gains across the majority of these evaluation dimensions.

**Efficient Beam Search with Probability.** To support real-world deployment, a search strategy that is both efficient and reliable is critical. In Table 2, we compare our proposed probabilistic beam search against a straightforward Process Reward Model (PRM) strategy, which greedily selects the top-1 token at each step based solely on reward scores.

Our findings reveal that current autoregressive world models—much like LLMs and LMMs—often suffer from unstable and inconsistent decoding paths. As a result, incorporating a verification mechanism that can identify and follow more reliable generation trajectories is crucial. Our probabilistic beam search strikes a strong balance between performance and robustness, achieving better results without incurring additional overhead.

| Model | N | Alg. | FVD | FID | IQ | MS | VQ | TC | DD | FC |
|---|---|---|---|---|---|---|---|---|---|---|
| 4B | 1 | - | 637.08/120.30 | 67.75/10.58 | 63.48 | 0.9822 | 3.86 | 3.56 | 3.86 | 3.68 |
| | 4 | PRM | 614.00/116.51 | 52.68/11.48 | 63.79 | 0.9836 | 3.68 | 3.38 | 3.68 | 3.48 |
| | | Ours | 612.68/114.27 | 50.39/10.35 | 64.39 | 0.9837 | 3.86 | 3.56 | 3.85 | 3.67 |
| | 16 | PRM | 616.89/121.07 | 47.29/10.33 | 64.87 | 0.9844 | 3.89 | 3.58 | 3.88 | 3.71 |
| | | Ours | 590.34/120.48 | 43.69/10.27 | 64.98 | 0.9846 | 3.92 | 3.64 | 3.90 | 3.74 |
| 12B | 1 | - | 560.86/117.23 | 67.10/10.67 | 63.73 | 0.9807 | 3.94 | 3.63 | 3.93 | 3.76 |

Table 2: Comparison between COSMOS-4B and COSMOS-12B under different search algorithms. $N$ is sample number. For our beam search with probability, $M$ is set as sqrt($N$).

We also investigate the text+image-to-video modality in Table 3 and the findings are aligned with those on the image-to-video modality.

| Model | N | Alg. | FVD | FID | IQ | MS | VQ | TC | DD | FC |
|---|---|---|---|---|---|---|---|---|---|---|
| 5B | 1 | - | 728.68/126.63 | 59.71/10.47 | 63.48 | 0.9822 | 3.80 | 3.44 | 3.81 | 3.63 |
| | 2 | ORM | 659.79/111.96 | 59.45/10.01 | 63.90 | 0.9840 | 3.89 | 3.55 | 3.88 | 3.71 |
| | 4 | PRM | 641.5/112.15 | 52.60/10.07 | 64.39 | 0.9843 | 3.89 | 3.55 | 3.88 | 3.71 |
| | | Ours | 628.01/110.02 | 50.16/9.98 | 64.77 | 0.9848 | 3.91 | 3.57 | 3.92 | 3.73 |
| 13B | 1 | - | 616.92/109.77 | 59.71/11.48 | 63.79 | 0.9834 | 3.92 | 3.55 | 3.94 | 3.75 |

Table 3: Comparison between COSMOS-5B and COSMOS-13B under different search algorithms. $N$ is sample number. For our beam search with probability, $M$ is set as sqrt($N$).

**Human Evaluation.** To better demonstrate the performance gain over larger pretrained model, we opt for human evaluation for more faithful analysis. Following recent advances Bar-Tal et al. (2024); Blattmann et al. (2023a); Chen et al. (2023), we adopt the Two-Alternative

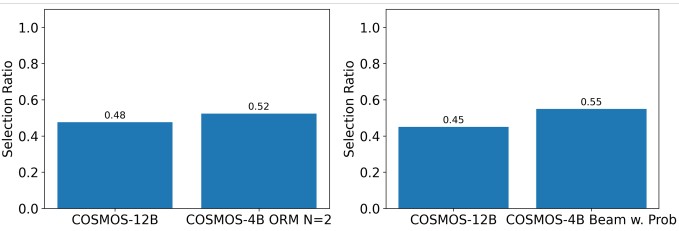

Figure 8: COSMOS-4B with test-time scaling vs. COSMOS-12B.

Forced Choice (2AFC) protocol. Specifically, participants are shown side-by-side video pairs and asked to select the one they find better. We collect a total of 3,685 responses from 24 participants. As shown in Figure 8, outputs from the smaller model enhanced with test-time scaling are often preferred over those from the larger baseline. While this study focuses on

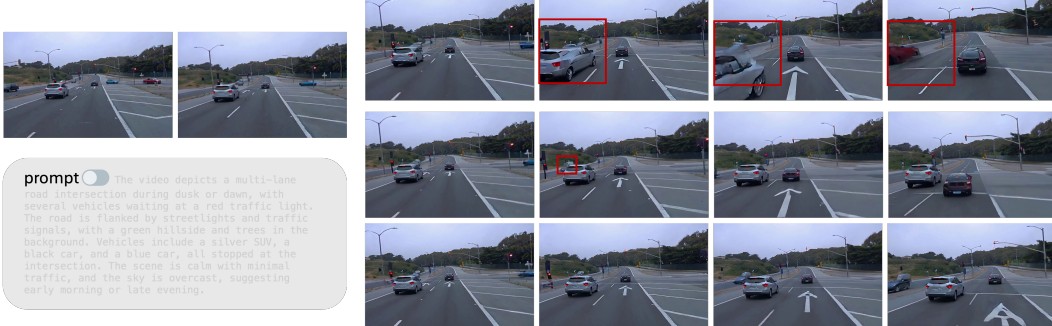

Figure 9: COSMOS-4B without (top) and with ORM (middle) and our (bottom) test-time scaling.

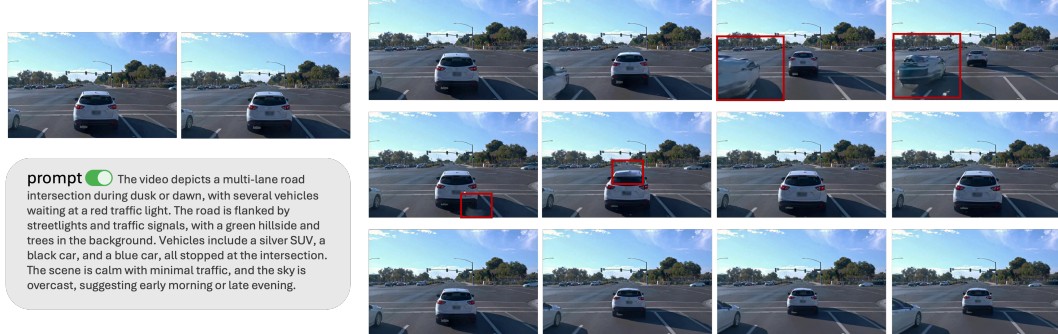

Figure 10: COSMOS-5B without (top) and with ORM (middle) and our (bottom) test-time scaling.

scaling smaller models, we note that applying test-time scaling to larger models can further push performance toward the frontier—an exciting direction we leave for future work.

## 5.2 Qualitative Results

We present qualitative examples in Figure 9 and Figure 10. Each example includes video outputs under different test-time scaling strategies. The baseline model frequently produces inconsistent objects, unstable backgrounds, or unnatural appearances. In contrast, test-time scaling consistently mitigates these issues—enhancing temporal consistency and improving overall visual fidelity.

## 6 Conclusion

In this work, we presented **SWIFT**, the first test-time scaling framework specifically designed for world foundation models (WFMs). Addressing the high computational cost and data limitations of training and scaling WFMs, SWIFT offers an efficient alternative by reallocating computation during inference. Our framework combines a modular evaluation toolkit with process-level test-time strategies, including fast tokenization, probability-based Top-K pruning, and beam search.

Empirical results on the COSMOS model demonstrate that test-time scaling not only improves output quality, but does so in a compute-optimal manner—allowing smaller models to match or even outperform larger ones under the same compute budget. These findings establish that test-time scaling laws hold for WFMs and open up a practical, scalable pathway for deploying WFMs more efficiently, without retraining or model enlargement.

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

## A    More Related Works

**World Models and Video Generative Models**    World models Ha & Schmidhuber (2018) aim to learn compact representations of the real world from sensory data, enabling reliable prediction of future states. Traditional physics-based models Watters et al. (2017) excel in controlled, low-dimensional environments but struggle with generalization across diverse scenarios. Recent advances Hafner et al. (2019); Okada & Taniguchi (2022) leverage deep learning to construct world models directly from visual inputs, often employing recurrent architectures or generative approaches to capture dynamic real-world processes.

Video generative models have progressed from early low-resolution systems to modern frameworks that synthesize high-quality, realistic videos. Two primary approaches have emerged: diffusion-based models Ho et al. (2022); Blattmann et al. (2023b), known for their superior visual quality, and autoregressive models Deng et al. (2024); Jin et al. (2024), which effectively capture temporal dependencies by generating frames sequentially. While diffusion methods excel in visual fidelity, autoregressive techniques benefit from advances in language modeling and offer improved integration of complex reasoning. Our study leverages an autoregressive approach within world foundation models, demonstrating that scaling test-time computation can enhance verification and overall performance in autonomous driving contexts.

**Scaling Test-Time Computation.** Inspired by how humans allocate more cognitive effort to tackle complex problems, recent work has explored scaling test-time computation to improve the performance of Large Language Models (LLMs) across domains such as mathematical reasoning Wang et al. (2023); Zhang et al. (2025); Xin et al. (2024); Sun et al. (2024), code generation DeLorenzo et al. (2024); Ni et al. (2023); Zhu et al. (2024), and multi-step planning Zhang et al. (2024); Gal et al. (2024); Xue et al. (2024). One class of approaches enhances the input side, leveraging techniques like Chain-of-Thought (CoT) prompting Wei et al. (2022); Kojima et al. (2022) to guide models through more structured reasoning. Others focus on the output side, introducing strategies such as self-consistency Wang & Zhou (2024), iterative decoding, or verifier-based selection Cobbe et al. (2021); Lightman et al. (2023); Snell et al. (2024) to improve response quality. Verifier-based methods, in particular, have proven to be robust and general-purpose. Outcome Reward Models (ORMs) evaluate final outputs to select the best among multiple candidates Cobbe et al. (2021), while Process Reward Models (PRMs) provide finer-grained feedback during generation by assessing intermediate reasoning steps Lightman et al. (2023); Ma et al. (2023). Recent work has shown that increasing inference-time compute—rather than scaling model size—can yield stronger gains Snell et al. (2024). Notably, OpenAI's o1 model OpenAI demonstrates the effectiveness of such techniques across complex reasoning tasks. Building on these insights, we investigate whether verifier-based test-time scaling strategies can be adapted to enhance autoregressive video generation in world foundation models, where sequential consistency and high-fidelity synthesis are critical.

