# OpenReview forum: "Can Test-Time Scaling Improve World Foundation Model?"
_colmweb.org/COLM/2025/Conference — COLM 2025_

### Official Review · Reviewer_vAmQ · 2025-04-14

**Rating:** 7
**Confidence:** 4
**Ethics Flag:** 1

**Summary:**

This paper discusses test-time scaling for video-generating world foundational models. A retraining-free test-time scaling framework, **SWIFT**, is proposed, which features: **(1)** alignment with goals of physical realism and consistency, and **(2)** a test-time inference strategy supporting CoT, ToT, etc., tailored for WFMs which typically use diffusion decoders.

**The evaluation metric includes**: (1) 3D consistency evaluated with 3D foundational models; (2) Temporal Consistency evaluated with CLIP, DINO similarity scores; (3) Spatial Relationship Awareness, evaluated with ; (4) Perceptual Quality, assessed with LAION aesthetic predictor; and (5) Test-to-Video Alignment, evaluated with ClipScore and X-ClipScore; and this framework is extendable to other metrics.

**For the Inference-time scaling method**, the authors discussed the reward choices, the action designs, and the inference-time search algorithm.

**For reward choices**, they compare preference-based rewards (i.e., VideoScore) with Rule-based rewards for visual quality and temporal consistency (i.e., aesthetic quality and object permanence) on visual quality and temporal consistency on the task of autonomous driving.

**For action designs**, they discuss why they use test-time search to explore inference-time scaling laws.

**For Search Algorithms**, they (1) validate the usefulness of inference-time scaling with BoN methods; and (2) find that WFM's discrete-token decoder is a lightweight proxy to diffusion-based heavy decoders; (3) use Probability-based Top-k pruning as a improved BoN method and (4) use beam-search to maintain efficiency.

**For experimental validation**, the authors conduct experiments based on driving scenarios in nuScenes and Waymo, evaluating FVD, FID and further metrics (including VQ, TC, DD, FC, MS, IQ, human evaluation, etc. ). The authors (1) conduct reward ablation to choose Temporal Consistency + Perceptional Quality as scaling reward; (2) show conceptually that Inference-time scaling can improve performance for small models with BoN; and (3) their proposed probabilistic beam search strikes balance between performance and robustness.

**Questions To Authors:**

When it comes to comparing Rule-based rewards and Preference-based rewards (around Figure 3), the rule-based rewards are chosen as visual quality and temporal consistency scores, which I think is closer to the evaluation metric (i.e. visual quality and consistency here)? Hence is it natural that VideoScore-based reward cannot outperform rule-based rewards, since the output is evaluated with these rules? While for the FVD metric you use in Figure 7, the output is not evaluated with these rules, so perhaps Preference-based rewards (e.g. VideoScore-based reward) can out-perform rule-based rewards? Could you conduct experiments with preference-based rewards to validate (or diss-validate) this?

I appreciate this work in general; I'm not asking the authors to conduct all experiments I mentioned (or implied) in weakness and questions section: some of these experiments are indeed very resource-consuming to carry out.

**Reasons To Accept:**

The paper presents a **timely discussion** about improving performance of World Foundation Models, especially the video models, with Test-time scaling. The paper **validates the idea** that with test-time designs (i.e., reward designs, search algorithm designs, etc.), we can use smaller amount of compute to improve performance: small models + test-time inference can be compute-optimal compared to larger models, for World Foundation Models.

The evaluation framework proposed is **extendable**, and could be of good impact to the community to evaluate further models potentially.

The methods proposed looks **good and reasonable**, and comparisons with baselines and ablation studies indeed show that the authors' methods can improve performance while save cost.

The paper is **well written** with clear presentations about its methods, experiment designs and results.

**Reasons To Reject:**

This work is mainly based on one series of WFM (i.e., the COSMOS-4B and COSMOS-12B models). Potential discussions about other WFMs could make this work even sounder.

The fast Tokenizer decoder is provided by the COSMOS work, and the intuition that using a tokenizer decoder is faster than diffusion decoder, is straightforward.

A minor point: the evaluation metrics and discussions proposed are mainly for World Foundation Models which generate videos. Of-course this is very important and timely since many WFMs currently generate videos, but potential discussions about WFMs in other modalities (e.g., 3D meshes, 3D videos?) could be even more inspiring.

---

> ### Author Response · Authors · 2025-06-03
>
> Thank you for acknowledging our work “timely discussion of WFMs”  with an “extendable” evaluation framework. We sincerely appreciate the detailed feedback and have addressed your comments as follows.
>
> ### Q1: Generalization to Other WFMs
> **A1:** COSMOS represents the recent SOTA world foundation models, trained on over 10⁸ video clips with 10,000 H100 GPUs over three months. It goes beyond single-task domains like autonomous driving, making it a strong and representative model series for our study. Furthermore, COSMOS utilizes a **standard autoregressive transformer architecture**, a common backbone in many large-scale models. Precedent in LLM research indicates that test-time scaling principles generalize across different model families sharing similar foundational architectures. Consequently, the methods we investigate (e.g., test-time sampling, verifier designs) are inherently model-agnostic and readily extendable to other WFMs built upon the same autoregressive transformer paradigm.
>
> ---
>
> ### Q2: Fast Tokenizer
> **A2:** We agree that the tokenizer decoder originates from the COSMOS work. However, it is typically not used in COSMOS for final output due to lower quality (e.g., artifacts, blurriness); instead, COSMOS relies on a much slower diffusion decoder for high-quality generation.
> Our key insight is that **verifiers in TTS only need to reliably distinguish between better and worse candidates, and do not require to give a precise score.** This insight drives us to use this fast tokenizer decoder into SWIFT's search and pruning stages to make TTS practical, as diffusion decoding is over 9000× slower.
> We also prove the strong correlation between scores derived from the tokenizer decoder and those from the diffusion decoder in Figure 5. This confirms that the tokenizer decoder can serve as a reliable and highly efficient proxy for candidate evaluation during TTS.
>
> ---
>
> ### Q3: Potential for WFMs in Other Modalities (e.g., 3D)
> **A3:** Thank you for the insightful comment. We agree that extending evaluation and discussion to other modalities such as 3D meshes and 3D videos is a promising direction. In fact, our evaluation toolkit was designed with this in mind, including metrics such as **3D consistency and spatial relationship awareness**.
> While current WFMs like COSMOS primarily generate videos and do not directly produce 3D meshes or scenes, we took a step toward broader applicability by evaluating **3D video generation**, following the protocol in the COSMOS paper. Specifically, we tested COSMOS-4B and COSMOS-4B+best-of-2 on 500 videos from RealEstate10K test set, using rewards including temporal consistency, perceptual quality, and 3D consistency (Due to time limit, this evaluation focused on COSMOS-4B, and we omitted spatial relationship metrics in this case since no text prompts were used). As shown in the accompanying table, our test-time scaling approach significantly improved performance in this 3D-aware setting. We will incorporate a brief summary of these findings in the revised discussion section.
>
> | | FVD   | FID   | Sampson error  ↓     | Pose estimation success rate (%)  ↑  |
> |---------|-------|-------|--------|------|
> | 4B      | 48.60 | 10.16 | 0.2532 | 46.1 |
> | 4B+best-of-2   | 47.53 | 9.94  | 0.2431 | 53.4 |
>
> ---
>
> ### Q4: Comparison of Rule-Based and Preference-Based Rewards
> **A4:** Thank you for this insightful question. To ensure a fair comparison in Figure 3, we used the same aspects (visual quality, temporal consistency) as reward for both Rule-Based and Preference-Based Rewards. The key difference is in how these rewards are derived:
> - Rule-based rewards are computed from feature statistics (e.g.,  DINO similarity, X-CLIPScore),
> - Preference-based rewards are learned via regression from human-annotated video preference pairs (e.g., using the VideoScore model).
>
> Moreover, to avoid reward hacking, the evaluation metrics, geometric mean of FVD, FID, IQ(imaging quality), MS(motion smoothness), have no overlap with reward functions used for TTS.
>
> Regarding your hypothesis about FVD: we conducted ablations focusing on FVD and FID in the following table. While preference-based rewards occasionally achieved slightly better FVD, this often coincided with notably worse FID. Overall, our rule-based approach yielded a more consistent and favorable balance across these metrics, supporting our selection of rule-based reward in our study.
>
> - `n`: n in Best-of-n
> - `R:`: Rule-Based Reward
> - `P:`: Preference-Based Reward
> - `G`: Geometric mean of normalized FVD and FID  ↑
> | n  | R:FVD/FID | R:G | P:FVD/FID | P:G |
> |----|-----------|-----|-----------|-----|
> | 1  | 637.08/67.75 | 0.000 | 637.08/67.75 | 0.000 |
> | 2  | 622.82/58.93 | 0.054 | 619.06/62.09 | 0.049 |
> | 4  | 613.77/52.01 | 0.092 | 628.69/59.19 | 0.041 |
> | 8  | 599.10/49.27 | 0.128 | 602.98/56.50 | 0.094 |
> | 16 | 599.08/45.75 | 0.139 | 585.11/54.99 | 0.124 |
> | 32 | 596.38/42.48 | 0.154 | 584.36/54.69 | 0.126 |

---

> > ### Comment · Reviewer_vAmQ · 2025-06-03
> >
> > I would like to thank the authors for providing more details and conducting supplementary experiments. Most of my concers have been solved, hence I would like to increase my score from a 6 to a 7.

---

> > > ### Author Response · Authors · 2025-06-03
> > >
> > > Dear Reviewer vAmQ,
> > >
> > > Thank you very much for your thoughtful feedback and for recognizing our work as a **good paper to accept**.
> > >
> > > We’re especially grateful that you described our work as a **timely discussion** and highlighted the **extensibility of our evaluation framework** as a valuable resource for the broader community. We also appreciate your acknowledgment of the **practical impact** of our approach, particularly the finding that **small models with test-time scaling, can achieve compute-optimal performance and even beat larger models**.
> > >
> > > We're glad that our rebuttal and supplementary experiments effectively addressed your concerns, and we sincerely appreciate you updated the evaluation from 6 to 7. Your encouraging assessment underscores its potential impact. Thank you again for your insightful and constructive review.

---

### Official Review · Reviewer_oo4B · 2025-05-12

**Rating:** 7
**Confidence:** 3
**Ethics Flag:** 1

**Summary:**

The paper introduces SWIFT, a test-time scaling framework specifically designed for World Foundation Models (WFMs), which are models that simulate the physical world by predicting future states from current observations and inputs. The authors address the challenge that WFMs require substantial computational resources for pretraining and are constrained by available data during post-training. Instead of building larger models or retraining, they propose using additional computation at inference time (test-time scaling) to improve performance.

**Questions To Authors:**

The temperature parameter τ in your probability-based Top-K pruning seems important for balancing exploration and exploitation. Could you elaborate on how sensitive your results are to this parameter and how you selected its value?

**Reasons To Accept:**

1/ Addresses a significant practical challenge in WFMs - the high computational cost of training and inference

2/ Provides a compelling alternative to simply building larger models, showing smaller models can achieve comparable performance with appropriate inference strategies
3/ Introduces a much-needed evaluation toolkit specifically designed for WFMs, filling a gap where existing video generation benchmarks don't align with physical modeling goals

4/ The fast tokenizer approach intelligently avoids running expensive diffusion decoding at every step

5/ Demonstrates compute-optimal scaling, where a 4B model with test-time scaling can match a 12B/13B model's performance

6/ Includes both rigorous quantitative metrics and human evaluation to validate the approach

**Reasons To Reject:**

While the framework shows promising results, there's limited ablation of hyperparameters within SWIFT (such as the temperature parameter in probability-based selection)

The human evaluation includes only 12 participants, which may be insufficient given the complexity of video quality assessment


The test-time scaling approach still requires multiple inference passes, which may present deployment challenges in resource-constrained environments

---

> ### Author Response · Authors · 2025-06-03
> **Rebuttal to Reviewer oo4B**
>
> Thank you for acknowledging our work “addresses significant practical challenges” and provides “much-needed evaluation toolkit”. We sincerely appreciate the detailed feedback and have addressed your comments as follows:
>
> ---
>
> ### Q1: Ablation of Temperature in SoftMax of Top-K Pruning
> **A1:** Thank you for highlighting this. In our framework, the temperature τ in the softmax-based Top-K pruning is **not a key hyperparameter**. We heuristically select 0.1 since we find the reward scores are close in magnitude. So a smaller τ helps sharpen the distribution to better differentiate the probabilities to prune out unsatisfactory paths.
> Following your suggestion, we conducted an ablation study by sweeping τ ∈ {0.01, 0.1, 0.5, 1.0} using our probability-based beam search (N = 4) on a nuScenes subset. We found that slightly varying t in range [0.1, 0.5] does not change the performance much.
>
> | Model                     | Temperature | FVD↓   | FID↓   |
> |--------------------------|-------------|--------|--------|
> | 4B                       | -           |637.08 | 67.75  |
> | 4B+ Beam w. Prob N=4     | 0.01        |   631.29 | 61.21  |
> |                          | 0.1         |  612.68 | 50.39  |
> |                          | 0.5         |    612.63     |  53.09      |
> |                          | 1           | 627.23 | 59.02  |
>
> We will include this ablation and other hyperparameter studies in the full revision for completeness.
>
> ---
>
> ### Q2: Human Evaluation
> **A2:** Thank you for the valuable feedback. To address your concern, we conducted an additional human study with **another 12 new independent participants**. Despite natural variation, the results remained consistent with our initial study:
> - The 4B model with test-time scaling continued to outperform the 12B baseline, with preferences increasing by ~5% (ORM) and ~8% (probability-based beam search).
> - Beam search remained favored over best-of-N ORM sampling.
>
> Due to the time limitation, we only added 12 participants to the rebuttal. For future revisions, we are committed to further expanding participant numbers for human studies.
>
> | Model                   | Selection Ratio |
> |------------------------|-----------------|
> | COSMOS-4B ORM N=2      |       52.4%        |
> | COSMOS-12B             |       47.6%      |
>
> | Model                   | Selection Ratio |
> |------------------------|-----------------|
> | COSMOS-4B Beam w. Prob | 54.3%             |
> | COSMOS-12B             | 45.7%             |
>
> ---
>
> ### Q3: Deployment in Resource-Constrained Case
> **A3:** We agree that test-time scaling (TTS) involves multiple inference passes. However, we respectfully argue that deployment cost should be considered alongside the training cost of larger models. Even from a deployment-only perspective, we observe a **compute-optimality** finding that multiple inferences with a small model can beat a single inference with a large model in terms of quality with similar cost. Similar trends have also been reported in the LLM literature [1]. Moreover, **parallel inference can alleviate runtime concerns**: because autoregressive models are typically memory‑bound rather than compute‑bound, it is feasible in practical systems to decode multiple paths concurrently.
>
> We emphasize that test-time scaling (TTS) is fundamentally a method for trading inference compute for improved performance. **One key motivation for studying TTS is to explore whether we can enhance a base model’s capability without any additional training or scaling.** Our results show that small models with TTS can beat larger models, offering a practical alternative when training/assessing a larger model is infeasible. This is particularly valuable in two scenarios:
> - When large models are unavailable—for example, the current 13B COSMOS model is not the upper limit by design, but a result of limited data and the high cost of WFM training;
> - When large models are closed-source.
>
> In such settings, **computation is not the primary constraint**. TTS offers a flexible, training-free pathway to scale performance under these limitations.
>
> ---
>
> [1] Scaling llm test-time compute optimally can be more effective than scaling model parameters. arXiv preprint arXiv:2408.03314.

---

> > ### Author Response · Authors · 2025-06-08
> > **Follow-Up on Rebuttal Response**
> >
> > Dear Review oo4B:
> >
> > Thank you again for the time and effort you’ve dedicated to reviewing our work! Your insightful comments have been instrumental in helping us improve the quality and clarity of our paper!
> >
> > We have carefully addressed all of your comments in our response above. To briefly summarize:
> >
> > - We added an ablation study on the temperature parameter τ in the softmax-based Top-K pruning and found the performance remains **robust across the range [0.1, 0.5]**.
> >
> > - We conducted an additional human study involving **12 new independent participants**. While there is natural variation, the overall performance trends remain consistent with our previous findings.
> >
> > - We agree that TTS involves multiple inference passes. However, our findings show that **small models with TTS can outperform larger models with simiar cost**, a trend also observed in the LLM literature. TTS provides a training-free and flexible approach to **scaling performance**—particularly valuable when **large models are inaccessible** or **closed-source**.
> >
> > As the discussion period deadline approaches, we would be grateful if you could review our responses. We will be online over the next few days and are happy to address any further questions or concerns you may have. Thanks!

---

> > ### Author Response · Authors · 2025-06-09
> > **Friendly Reminder as the Rebuttal Phase Nears Completion**
> >
> > Dear Reviewer oo4B,
> >
> > Thank you again for your thoughtful and constructive feedback during the review phase. As the rebuttal period draws to a close, we would greatly appreciate it if you could take a moment to review our responses. We believe they addressed your concerns and clarified the key points of our work, **the first general and extensible pipeline to study test-time scaling for emerging world foundation models**, encompassing (i) a curated testbed, (ii) a modular and scalable multimodal evaluation toolkit, and (iii) inference-time strategy exploration for WFMs.
> >
> > We truly appreciate your time and effort in supporting the review process.
> >
> > Warm regards,
> >
> > Authors of Paper 59

---

> > > ### Author Response · Authors · 2025-06-10
> > > **Friendly Reminder as the Rebuttal Phase Complete in one Day**
> > >
> > > Dear Reviewer oo4B,
> > >
> > > Thank you again for your thoughtful and constructive feedback during the review phase. As the rebuttal period draws to a close in **1 day**, we would greatly appreciate it if you could take a moment to review our responses. We remain available online and would be happy to clarify anything further if needed. We sincerely appreciate your efforts in helping us improve the manuscript.
> > >
> > > Warm regards,
> > >
> > > Authors of Paper 59

---

> ### Author Response · Authors · 2025-06-06
> **Follow-Up on Rebuttal Response**
>
> Dear  Reviewer oo4B:
>
> We sincerely appreciate the time and effort you have dedicated to reviewing our work! Your insightful suggestions have been invaluable in improving the quality of our work. We have carefully considered all your feedback and have addressed all your comments in our response above.
>
> As the discussion period deadline approaches, we would be grateful if you could review our responses. We will be online over the next few days and are happy to address any further questions or concerns you may have. Thanks!

---

### Official Review · Reviewer_qtYN · 2025-05-13

**Rating:** 5
**Confidence:** 4
**Ethics Flag:** 1

**Summary:**

The paper studies test-time compute in the context of world foundation models. In particular, they (1) observe and quantify improvements in generation quality (including things like physical consistency, perception etc) of world foundation models with difference inference compute strategies (e.g. best of N, beam search). (2) They study reward design in detail, formulating video generation as an MDP.

Overall this is a nice paper with interesting insights (e.g., it was interesting to see different process rewards at play -- it wasn't obvious which ones would be most effective). However, I am not entirely sure the paper is within scope of COLM. While the COLM call for papers is not entirely restricted to _language_ models (it does mention applications to multimodality and other domains), the current paper is entirely focused, as far as I can tell, towards video models, and is perhaps more suitable for a less LLM-specialized venue?

**Reasons To Accept:**

- Studies an interesting problem -- inference time scaling quality for video generation models. This has practical impact, as video gen models are costly to train, and one could simple apply more inference compute to smaller models for better quality.
- Good empirical results -- e.g., shows evidence that test-time scaling laws apply to WFMs, showing that a 4B parameter model with test-time scaling can match or outperform a 13B model while using similar compute.
- Solid technical contribution along various axes -- presents a system with multiple components (fast tokenizer, probability-based pruning, efficient beam search) designed specifically to make WFM inference efficient. Additionally, it was nice to see the effect of various rewards specific to video gen.

**Reasons To Reject:**

- The paper doesn't seem very in scope for the venue, focusing entirely on video gen models.
- Some of the emphasized results feel a bit obvious -- e.g., "Observation 1" that test-time scaling and improvements exist for WFMs seems fairly intuitive.
- It's unclear how generalizable the results are, since the authors studies one large WFM (given compute constraints). E.g., the results on compute-optimality at small N, motivating a best-of-2 strategy -- how general could this be?
- Some sentence structure and grammar issues, e.g. "similar to the Process Reward Model (PRM) in the LLM Ma et al. (2023)"

My primary concern remains appropriateness for the venue, with other concerns being significant but secondary.

---

> ### Author Response · Authors · 2025-06-03
> **Rebuttal to Reviewer qtYN**
>
> Thank you for acknowledging our work’s “interesting insights” on “test‑time computation of world foundation models.” We have addressed your concerns below.
>
> ---
>
> ### Q1: Regarding the Scope and Venue Fit
> **A1:** Thank you for the thoughtful feedback. We believe our work aligns well with COLM’s scope:
>
> - **Multimodal foundation model, not just a video generation model:** COSMOS itself is a multimodal model that models the physical world with both human-provided descriptions (text domain) and visual inputs (image or video domain). Video generation is the evaluation protocol to assess a WFM's ability to model the complex physical world, which does not mean we are studying a video generation model. We will revise our writing to avoid this confusion.
> - **LLM-inspired modeling:** COSMOS is an autoregressive model that predicts the physical world via next-token prediction, the paradigm inspired by the success of LLMs. We believe it is very relevant to the new application of LLMs, aligning with Call for Papers #18 (LMs on diverse domains and novel applications).
> - **Aligned with COLM’s multimodal and world modeling tracks:** COLM Calls for Papers #14 and #13 explicitly invite work on Multimodal LMs and LMs and the World, respectively, which our contributions directly fit, making us choose COLM.
> - **Precedent multimodality foundation model works in COLM:** COLM has recently accepted several multimodal works [1–5], including on text-to-image generation [2,3], adversarial image reasoning [1], and compositional visual tasks [4,5].
>
> ---
>
> ### Q2: Regarding the Obviousness of Observation
> **A2:** We would like to argue that it is an open question (non-obvious) whether similar test-time scaling (TTS) trends hold for world foundation models (WFMs) as LLMs [6] and text-to-image generation [7], because (1) WFMs solve a hard task that models the physical world and involves more modalities; (2) Unlike LLMs, which often study tasks like math and puzzles with well-defined, rule-based rewards, WFMs need to operate under much weaker reward signals like visual fidelity and temporal consistency.
>
> In this paper, we bridge this gap by presenting the first systematic study of test‑time scaling in WFMs. **Though concluding that TTS exists may appear trivial from the experiments**, we rigorously validate it through non‑trivial efforts: evaluating multiple model sizes, varying N in best‑of‑N sampling, and collecting thousands of video clips. The first WFM evaluation toolkit is established to fill the assessment gap, and practical TTS strategies are explored to reach a conclusion.
>
> ---
>
> ### Q3: Generalization of Results
> **A3:** **Why COSMOS?**
> COSMOS represents the recent SOTA world foundation models, trained on over 10⁸ video clips with 10,000 H100 GPUs over three months. It goes beyond single-task domains like autonomous driving, making it a strong and representative model series for our study. To the best of our knowledge, COSMOS was the only publicly accessible and fundamental WFM available at the time of our work.  ***If other suitable WFMs are suggested, we are happy to include them.***
>
> Furthermore, COSMOS utilizes a standard autoregressive transformer architecture, a common backbone in many large-scale models. Precedent in LLM research indicates that test-time scaling principles generalize across different model families sharing similar foundational architectures. Consequently, the methods we investigate (e.g., test-time sampling, verifier designs) are inherently model-agnostic and readily extendable to other WFMs built upon the same autoregressive transformer paradigm.
>
> To bolster the robustness and generalizability of our findings concerning scaling laws, we conducted a comprehensive evaluation:
>
> - Evaluate across all available model sizes in the autoregressive COSMOS family (4B, 5B, 12B, 13B)
> - Sweep N from 1 to 32 in best-of-N settings
> - Compare multiple verifier strategies, including ORM, PRM, and our proposed probability-based beam search.
>
> Regarding the "best-of-2" observation, we concur that the precisely compute-optimal N may vary on other WFMs. However, the critical insight is the consistent self-evolving trend: performance robustly improves with increasing N (as illustrated in Figure 3). This mirrors observations in LLM scaling literature, where general scaling laws hold broadly, even if specific numerical parameters differ across individual models.
>
> ---
>
> ### Q4: Typos
> **A4:** Thank you so much for pointing out the typos in our manuscript, which is important to further improve our quality. We have revised the paper.

---

> > ### Author Response · Authors · 2025-06-03
> > **References**
> >
> > ### References
> >
> > [1] Stop Reasoning! When Multimodal LLM with Chain-ofThought Reasoning Meets Adversarial Image. COLM 2024
> >
> > [2] Can MLLMs Perform Text-to-Image In-Context Learning? COLM 2024.
> >
> > [3] Commonsense-T2I Challenge: Can Text-to-Image Generation Models Understand Commonsense? COLM 2024.
> >
> > [4] IllusionVQA: A Challenging Optical Illusion Dataset for Vision Language Models. COLM 2024.
> >
> > [5] ExoViP: Step-by-step Verification and Exploration with Exoskeleton Modules for Compositional Visual Reasoning. COLM 2024.
> >
> > [6] Scaling llm test-time compute optimally can be more effective than scaling model parameters. arXiv preprint arXiv:2408.03314.
> >
> > [7] Can We Generate Images with CoT? Let's Verify and Reinforce Image Generation Step by Step. CVPR 2025.

---

> > ### Comment · Reviewer_qtYN · 2025-06-07
> >
> > Thanks to the authors for considering my review and their clarifications.
> >
> > I will leave it up to the ACs to judge whether the paper is in scope. I think I don't fully agree with some of the points brought up -- for example, WFMs presented here seem sufficiently different from "[...] several multimodal works [1–5]", but it is not a strongly-held view.
> >
> > I also don't see why "WFMs solve a hard task that models the physical world and involves more modalities" would be a reason for the _existence_ of TT scaling behaviour to be surprising (even if the precise form is different from LLMs etc). I therefore think that "performance robustly improves with increasing N (as illustrated in Figure 3). This mirrors observations in LLM scaling literature, where general scaling laws hold broadly" should be quite expected, and the paper doesn't present general arguments to explore this phenomenon (but empirically studies it on an available model). I'm happy to raise my score by one, as I understand that the authors cannot in practice test other WFMs to draw more general conclusions.

---

> > > ### Author Response · Authors · 2025-06-08
> > > **Further Response to Reviewer qtYN**
> > >
> > > Thank you so much for your thoughtful engagement and comments.
> > > We would like to offer additional clarification regarding your remaining concern about (1) **the scope of the venue**, as well as (2) your view that some of our findings may be **“quite expected” or “obvious”**. We hope our clarifications help address your concerns about the scope and support consideration for further raising the acceptance rating.
> > >
> > > **(1) Scope Alignment with COLM**.
> > >
> > > Thank you again for recognizing that our work studies an interesting problem and offers solid technical contributions and strong empirical results. We understand that your primary concern remains the paper’s appropriateness for the Conference on Language Modeling (COLM), as it may appear to focus primarily on language-related models.
> > >
> > > - Our work on test-time scaling in multimodal World Foundation Models (WFMs) aligns not only with “Inference algorithms for LMs” (#10), “LMs on diverse domains and novel applications” (#18), “Multimodal LMs” (#14), but also with “LMs and the world” (#13) and “LMs and embodiment” (#15), as COSMOS-like WFMs are fundamental to advancing autonomous driving—a key evaluated application in our paper. Our submission aligns explicitly with multiple directions outlined in the Call for Papers, which shaped our decision. ***We chose to submit our work to COLM because we found it well aligned with the themes and vision presented in the Call for Papers***.
> > >
> > > - Accepted papers in COLM, including [1–5, 8, 9], show ***COLM’s interest in evaluating and understanding multimodal generation models***, including text-to-image generation [2, 3, 9], adversarial image reasoning [1], compositional visual tasks [4, 5], and evaluation metric development [8]. ***World Foundation Models (WFMs), which model future world states as video from multimodal inputs such as text, images, and past video [10]***, operate in the same spirit.
> > >
> > > We hope this additional context, grounded in both the Call for Papers and recent COLM publications, helps address your remaining concern regarding scope.
> > >
> > >
> > > **(2) Why TTS in WFMs Is Nontrivial: Directly applying TTS in LLMs to models with more modalities may fail.**
> > > - We will cite the evidence: Prior work in text-to-image generation [7] has shown the failure case that directly applying Process Reward Models (PRMs) from LLMs offers minimal benefit, largely due to weaker and noisier reward signals (e.g., visual fidelity) compared to structured tasks like math or puzzles in LLMs. These challenges are further amplified in video generation, where temporal dynamics introduce additional complexity. As a result, ***assuming that TTS behavior from LLMs naturally generalizes to WFMs risks oversimplification***. Recent work in T2V[11] and T2I [7] also treats this as an open research question, underscoring the need for rigorous empirical investigation before drawing the conclusion that TTS exists in multimodal foundation models.
> > > - ***The witnessed failure of directly applying scaling in LLM to multimodality models motivates our technical contribution. In our revision, we will clarify the limitations of applying PRMs to T2I generation, to better frame our motivation and highlight why a dedicated, domain-specific study of TTS in WFMs, such as ours, is both timely and necessary.***

---

> > > ### Author Response · Authors · 2025-06-08
> > > **Continued Further Response to Reviewer qtYN**
> > >
> > > **(3) Our contribution goes beyond observing the TTS law—it lies in delivering the first general pipeline for studying and evaluating TTS in WFMs.**
> > >
> > > We respectfully wish to clarify that the apparent similarity between the observed TTS behavior in our work and that of LLMs may unintentionally understate the core contribution of our paper. Specifically, we introduce the ***first general and extensible pipeline for systematically evaluating*** test-time scaling in World Foundation Models (WFMs), encompassing (i) a curated testbed, (ii) a modular and scalable multimodal evaluation toolkit, and (iii) efficient inference-time strategy exploration for WFMs. ***We sincerely appreciate your interest in broader general arguments on other WFMs. In fact, our pipeline was intentionally designed with that goal in mind as a **first general and extensible pipeline**, and we believe it will serve as a strong foundation tool for evaluating future WFMs as more models become available to the community.***
> > >
> > > Prior to our work, several critical components were missing in this space:
> > > - No established benchmark testbed for **evaluating WFMs**;
> > > - No comprehensive evaluation toolkit tailored to the **multimodal and dynamic nature of WFM outputs**;
> > > - No **efficient or practical test-time scaling strategies** are adapted to the unique challenges of autoregressive WFM video generation.
> > >
> > > Our proposed pipeline addresses these limitations as follows:
> > > - **Testbed**: We construct a benchmark using real-world driving scenarios, selecting autonomous driving as a representative and high-impact WFM application domain.
> > >
> > >
> > > - **Evaluation Toolkit**: We develop a modular and extensible suite of evaluation metrics covering perceptual quality, temporal and 3D consistency, and instruction-following capabilities such as text-video alignment and spatial relationship awareness—key for multimodal world modeling.
> > >
> > >
> > > - **Test-Time Scaling Strategies**: We design practical and compute-efficient strategies tailored for autoregressive video generation, including fast tokenizer-based decoding, probability-based pruning, and probabilistic beam search. Notably, we explore the feasibility of using a lightweight, non-diffusion-based tokenizer during inference to accelerate verification, **motivated by the insight that test-time scaling primarily requires distinguishing good outputs from bad**, rather than producing arbitrarily high-fidelity scores.
> > >
> > > **References:**
> > >
> > > [8] What makes a good metric? Evaluating automatic metrics for text-to-image consistency, COLM 2024
> > >
> > > [9] Iteratively Prompting Multimodal LLMs to Reproduce Natural and AI-Generated Images, COLM 2024
> > >
> > > [10] https://developer.nvidia.com/blog/scale-synthetic-data-and-physical-ai-reasoning-with-nvidia-cosmos-world-foundation-models/?ncid=no-ncid
> > >
> > > [11] Video-T1: Test-Time Scaling for Video Generation, ArXiv 2025

---

> > > > ### Author Response · Authors · 2025-06-09
> > > > **Friendly Reminder as the Rebuttal Phase Nears Completion**
> > > >
> > > > Dear Reviewer qtYN,
> > > >
> > > > Thank you again for your thoughtful and constructive feedback during the review phase. As the rebuttal period draws to a close, we would greatly appreciate it if you could kindly take a moment to review our responses. We hope they further address your concerns.
> > > >
> > > > In particular, we would like to respectfully emphasize that drawing the test-time scaling (TTS) conclusion in WFMs is a non-trivial endeavor. Our work introduces the first general and extensible pipeline to systematically study TTS in emerging World Foundation Models (WFMs), comprising (i) a curated testbed, (ii) a modular and scalable multimodal evaluation toolkit, and (iii) inference-time strategy exploration tailored to WFMs. This pipeline lays the groundwork to validate and deepen our understanding of TTS behavior in multimodal WFMs, and—as you insightfully suggested—we believe it will be instrumental in discovering further insights as more general WFMs become available.
> > > >
> > > > We sincerely appreciate your time and effort in supporting the review process.
> > > >
> > > > Warm regards,
> > > >
> > > > Authors of Paper 59

---

> > > > > ### Author Response · Authors · 2025-06-10
> > > > > **Friendly Reminder as the Rebuttal Phase Complete in one Day**
> > > > >
> > > > > Dear Reviewer qtYN,
> > > > >
> > > > > Thank you again for your thoughtful and constructive feedback during the review phase. As the rebuttal period draws to a close in **1 day**, we would greatly appreciate it if you could take a moment to review our responses. We remain available online and would be happy to clarify anything further if needed. We sincerely appreciate your efforts in helping us improve the manuscript.
> > > > >
> > > > > Warm regards,
> > > > >
> > > > > Authors of Paper 59

---

> ### Author Response · Authors · 2025-06-06
> **Follow-Up on Rebuttal Response**
>
> Dear  Reviewer qtYN:
>
> We sincerely appreciate the time and effort you have dedicated to reviewing our work! Your insightful suggestions have been invaluable in improving the quality of our work. We have carefully considered all your feedback and have addressed all your comments in our response above, especially **why we believe our work aligns well with COLM’s scope**.
>
> As the discussion period deadline approaches, we would be grateful if you could review our responses. We will be online over the next few days and are happy to address any further questions or concerns you may have. Thanks!

---

### Decision · Program_Chairs · 2025-07-08

**Decision:**

Accept

**Comment:**

This paper provides an interesting illustration of how test-time scaling methods from LLM reasoning can be directly applied to world models to improve the performance and accuracy of world modeling. Reviewers indicated their excitement about the paper, and papers seems well suited to the CoLM community. Congratulations to the authors!